# Enhancing Visual Token Representations for Video Large Language Models via Training-free Spatial-Temporal Pooling and Gridding

**Bingjun Luo**[1], **Tony Wang**[1], **Hanqi Chen**[2], **Xinpeng Ding**[3]*

[1]Tsinghua University
[2]Shenzhen University
[3]Xidian University

bingjunluo@outlook.com, tonywang5454@gmail.com,
2550101019@mails.szu.edu.cn, xdingaf@connect.ust.hk

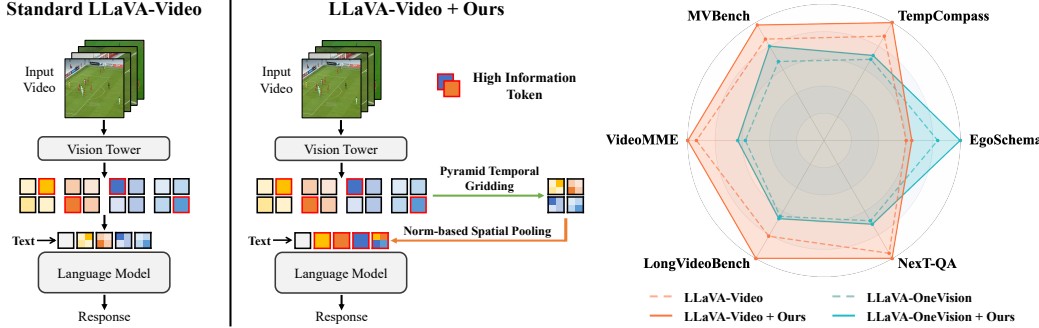

(a) Comparison of standard LLaVA-Video and our method.   (b) Performance improvement of ST-GridPool.

Figure 1: The proposed ST-GridPool enhances visual token representations while maintaining computational efficiency in a training-free manner, improving video understanding performance of widely adopted Video LLMs. Figure 1a illustrates the visual token construction of the standard Video LLMs and our method. Figure 1b presents the performance improvement of ST-GridPool on LLaVA-Video and LLaVA-OneVision model across various video understanding tasks.

## Abstract

Recent advances in Multimodal Large Language Models (MLLMs) have significantly advanced video understanding tasks, yet challenges remain in efficiently compressing visual tokens while preserving spatiotemporal interactions. Existing methods, such as LLaVA family, utilize simplistic pooling or interpolation techniques that overlook the intricate dynamics of visual tokens. To bridge this gap, we propose ST-GridPool, a novel training-free visual token enhancement method designed specifically for Video LLMs. Our approach integrates Pyramid Temporal Gridding (PTG), which captures multi-grained spatiotemporal interactions through hierarchical temporal gridding, and Norm-based Spatial Pooling (NSP), which preserves high-information visual regions by leveraging the correlation between token norms and semantic richness. Extensive experiments on various benchmarks demonstrate that ST-GridPool consistently enhances performance of Video LLMs without requiring costly retraining. Our method offers an efficient and plug-and-play solution for improving visual token representations. Our code is available in https://github.com/bingjunluo/ST-GridPool.

## 1 Introduction

Recent advances in Multimodal Large Language Models (MLLMs) have revolutionized multimodal understanding, delivering breakthroughs in image captioning, cross-modal retrieval, and video rea-

---

*Corresponding author.

soning (Radford et al., 2021; Li et al., 2023a; Liu et al., 2023; Zhang et al., 2024d). In these models, the quadratic complexity of self-attention mechanisms poses strict limitations on the input token length (Keles et al., 2023; Shao et al., 2025). However, video understanding inherently requires dense spatiotemporal analysis across thousands of visual tokens to capture precise motion dynamics and scene evolution (Xu et al., 2024a). Therefore, it is a pivotal challenge to strengthen visual token representations for better video understanding within the bounds of token length and computational limitations.

As shown in Figure 1, existing MLLMs, such as the LLaVA family (Liu et al., 2023; Li et al., 2024a; Zhang et al., 2024d), typically employ straightforward 2D pooling or interpolation to compress visual tokens into an appropriate shape. While these approaches are computationally lightweight, they often overlook the intricate spatiotemporal interactions inherent in visual tokens, leading to suboptimal performance. To address this problem, recent works like SF-LLaVA (Xu et al., 2024b) and TS-LLaVA (Qu et al., 2024) have introduced advanced training-free techniques to adapt Image LLMs for video understanding. These methods have demonstrated significant improvements on Image LLMs, even outperforming earlier Video LLMs (Xu et al., 2024b). However, over the past year, the rapid evolution of Video LLMs has resulted in remarkable advancements in video understanding, rendering mere optimizations on Image LLMs insufficient. There is an urgent need for training-free visual token enhancement strategies specifically for Video LLMs.

To address these challenges, we propose ST-GridPool, a novel training-free visual token enhancement method tailored for Video LLMs, which strategically refines visual tokens across both temporal and spatial dimensions while maintaining computational efficiency. Our method is composed of two key components: Pyramid Temporal Gridding and Norm-based Spatial Pooling. In **Pyramid Temporal Gridding** (PTG), we design a hierarchical gridding strategy over the temporal dimension, which grids and updates frame tokens from different segments of varying lengths. PTG enables multi-grained spatiotemporal feature extraction, capturing both short-term dynamics and long-term context without introducing additional trainable parameters. In **Norm-based Spatial Pooling** (NSP), we systematically explore the positive correlation between the token norm and the semantic richness of visual tokens, leveraging this insight to propose a norm-based 2D dynamic pooling approach. This approach preserves high-norm regions while adaptively compressing low-energy backgrounds, ensuring that high-information visual regions are prioritized and retained. By integrating this mechanism, we maximize the preservation of semantically meaningful visual details in the pooling process, significantly enhancing the representation power of the resulting visual tokens. The proposed ST-GridPool brings both components together to enhance visual token representations, achieving substantial improvements in video understanding tasks without requiring costly retraining or architectural modifications. This training-free paradigm offers a plug-and-play enhancement for existing Video LLMs like LLaVA-Video, demonstrating significant potential for video understanding applications. Our main contributions are summarized as follows:

- We propose the first training-free visual token enhancement method specifically designed for Video LLMs. By optimizing the visual token compression process, our approach significantly improves video understanding performance while maintaining computational efficiency.

- Our Pyramid Temporal Gridding introduces a hierarchical gridding strategy that captures multi-grained spatiotemporal interactions across varying temporal lengths. Our Norm-based Spatial Pooling leverages the positive correlation between token norms and semantic importance to effectively preserve high-value visual information during token compression.

- We conduct extensive experiments on 6 video understanding datasets using widely adopted Video LLMs, such as LLaVA-Video. Experimental results demonstrate that our method achieves consistent performance improvements across multiple models and diverse datasets.

## 2 RELATED WORK

### 2.1 VIDEO LARGE LANGUAGE MODELS

Video Large Language Models (Video LLMs) have emerged as a highly active research area on leveraging the powerful capabilities of large language models for video understanding. With the

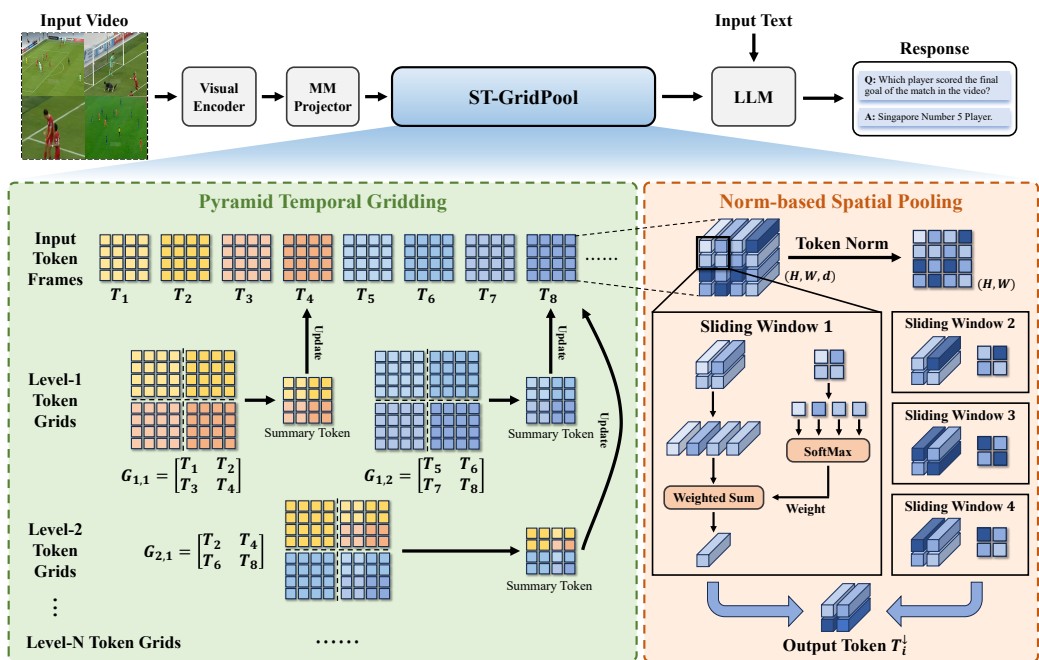

Figure 2: Overview of ST-GridPool. The method takes the token sequence $T_1, \cdots, T_N$ as input and outputs the pooled token $T_1^\downarrow, \cdots, T_N^\downarrow$, which consists of two main components: Pyramid Temporal Gridding and Norm-based Spatial Pooling.

rapid advancement of LLMs, a critical focus has emerged on harnessing their power for nuanced video analysis. Early efforts primarily adapt existing image-based LLM frameworks to the video field. VideoChat (Li et al., 2023b) develops an end-to-end chat-centric system by bridging video foundation models and LLMs via a learnable neural interface. PLLaVA (Xu et al., 2024a) adapts image-language pre-trained models for video tasks without additional parameters. Recent research focuses on further enhancing video LLMs from different aspects like cross-modal integration, long-context modeling, and efficiency optimization. LLaVA-OneVision (Li et al., 2024a) pioneers a single model architecture that unifies image, multi-image, and video understanding. mPLUG-Owl3 (Ye et al., 2024) introduces hyper attention blocks to efficiently process long image sequences. NVILA (Liu et al., 2024d) adopts a "scale-then-compress" approach to optimize both accuracy and efficiency in visual token processing. Based on existing Video LLMs, our approach aims at constructing efficient yet information-dense visual token representations from massive video data.

## 2.2 VISUAL TOKEN CONSTRUCTION FOR VIDEO LLM

Visual token construction serves as a critical bridge between raw video data and high-level semantic understanding in Video LLMs. Given the inherent redundancy of video content and the limited context windows of LLMs, extracting efficient yet informative visual tokens from long video sequences is essential for enabling efficient and accurate video understanding and reasoning. Current visual token construction techniques can be categorized into the following three types. First, single-frame feature extraction methods (Li et al., 2024a; Zhang et al., 2024d) leverage image-language models like CLIP to extract keyframe embeddings, followed by spatial downsampling via average pooling or bilinear interpolation. While computationally lightweight, these methods inherently neglect temporal dynamics and cross-frame dependencies. Second, temporal-aware modeling frameworks (Maaz et al., 2023; Xu et al., 2024a) integrate spatiotemporal attention layers to aggregate frame-level features, capturing motion patterns at the cost of quadratic complexity for long videos. Third, token compression strategies (Liu et al., 2024d) dynamically prune redundant visual tokens through learnable spatial-temporal aggregation. To summarize, most token construction optimizations require costly finetuning or model-specific architecture modifications. There remains a lack

of parameter-free visual token enhancement methods designed for Video LLMs which can improve token informativeness without altering model parameters or increasing inference latency.

## 3 METHOD

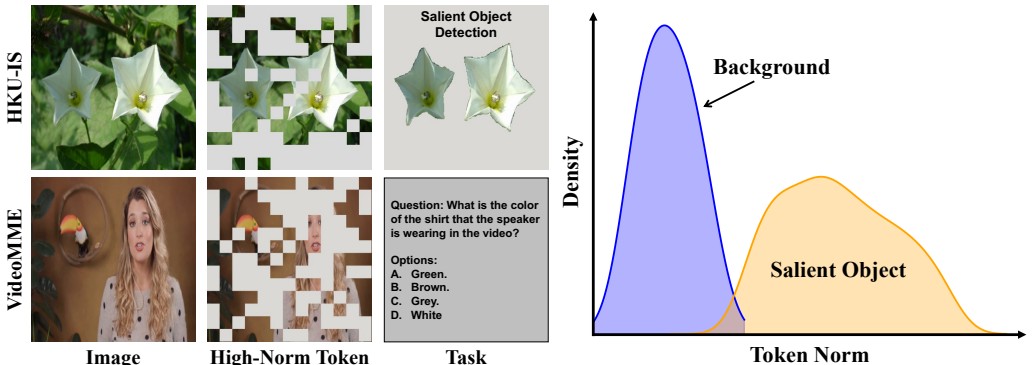

(a) Raw images, high-norm token and task of some samples.

(b) Token norm density distribution of the whole dataset.

Figure 3: Illustration of the visual token norm distribution discrepancy between salient object area and background area. (a) presents some samples with the raw image, the area of top 50% visual tokens in terms of $L_2$ norm, and the saliency groundtruth / question. (b) plots the density distribution of $L_2$ token norm among all validation samples from HKU-IS.

### 3.1 PROBLEM DEFINITION

Given an input video $\boldsymbol{V}$ with $R$ raw frames, conventional Video LLMs typically select $N \ll R$ frames $\boldsymbol{I}_1, \boldsymbol{I}_2, \cdots, \boldsymbol{I}_N$ from it via fixed-interval uniform sampling. The selected frames are then extracted by the vision tower $\Phi(\cdot)$, to generate frame tokens $\boldsymbol{T}_1, \boldsymbol{T}_2, \cdots, \boldsymbol{T}_N \in \mathbb{R}^{H \times W \times d}$ where $H \times W$ is the spatial dimension, $d$ is the feature dimension. In long-context scenarios like video understanding, the raw visual token length $N \times H \times W$ is usually excessive, posing heavy computational load and processing challenges for the downstream language model. Therefore, the raw visual tokens are further reduced by the downsampling function $\text{Down}(\cdot)$ to $\boldsymbol{T}_1^{\downarrow}, \boldsymbol{T}_2^{\downarrow}, \cdots, \boldsymbol{T}_N^{\downarrow}$. The down-sampled visual tokens are then fed into the language model to generate the final response. In this process, the downsampling function $\text{Down}(\cdot)$ is quite crucial for the visual token representations, as it must balance between reducing computational complexity and preserving critical spatiotemporal information, thereby achieving optimal video understanding performance.

### 3.2 METHOD OVERVIEW

As illustrated in Figure 2, the proposed ST-GridPool method consists of two components: Pyramid Temporal Gridding and Norm-based Spatial Pooling. Firstly, Pyramid Temporal Gridding (PTG) proposes a hierarchical gridding strategy to the temporal dimension, enabling multi-grained spatiotemporal feature extraction by combining and updating frame tokens from segments of varying lengths. Secondly, Norm-based Spatial Pooling (NSP) designs a norm-based 2D weighted pooling mechanism that prioritizes high-norm regions during spatial pooling to preserve rich visual semantics. Through the joint operation of these two components, ST-GridPool effectively enhances visual token representations in a training-free manner, achieving robust video understanding while maintaining computational efficiency.

### 3.3 PYRAMID TEMPORAL GRIDDING

Video LLMs like LLaVA-Video often adopt straightforward approaches by uniformly sampling and concatenating token sequences, implicitly assuming that temporal dynamics in videos are uniform

across scales. However, real-world video content frequently encompasses a spectrum of temporal granularities, ranging from rapid micro-motions (e.g., hand gestures) to gradual displacements (e.g., pedestrian walking), which demands a multi-scale temporal modeling strategy. To address this limitation, we propose the Pyramid Temporal Gridding (PTG), a hierarchical module that partitions the video sequence into multiple layers, each corresponding to distinct temporal segment lengths. PTG generates summary tokens at the end of each segment, enriching the temporal representation while maintaining computational efficiency. Given the original visual token sequence $\boldsymbol{T}_1, \boldsymbol{T}_2, \cdots, \boldsymbol{T}_N \in \mathbb{R}^{H \times W \times d}$, the details of this module are as follows.

As shown in Figure 2, PTG consists of $L$ levels, each corresponding to a specific segment length. For $l$-th level ($l = 1, 2, ..., L$), the segment length is defined as $K_l = K \cdot 2^{l-1}$, where $K$ is the base length of Level-1. The input visual tokens are divided into $N_l$ segments, where $N_l = \lceil N/K_l \rceil$. Therefore, the start frame index for the $j$-th segment in the $l$-th level is:

$$t_{l,j} = (j-1) \cdot K_l, \quad j = 1, 2, ..., N_l \tag{1}$$

Each segment spans the frame range $\{t_{l,j}, t_{l,j} + 1, ..., \min(t_{l,j} + K_l - 1, N - 1)\}$. For example, for an input token sequence with the length $N = 32$, we set the base length $K = 8$, and number of levels $L = 3$, the input sequence will be divided into 3 layers: For Level-1, the sequence is divided into 4 segments, each with $K_1 = 8$ frames; For Level-2, the sequence is divided into 2 segments, each with $K_2 = 16$ frames; For Level-3, the sequence is divided into 1 segment, encompassing all $K_3 = 32$ frames.

For the $j$-th segment at the $l$-th layer, a summary token is generated to capture the temporal dynamics. First, $m \times n$ frames are uniformly sampled from the segment. The sampling indices are:

$$\{t_{l,j} + k \cdot \lfloor K_l/(m \cdot n) \rfloor\}_{k=0}^{m \cdot n - 1} \tag{2}$$

The corresponding token grids of these frames are spatially concatenated to form an intermediate token grid $\mathbf{G}_{l,j}$:

$$\mathbf{G}_{l,j} = \begin{bmatrix} \mathbf{T}_{t_{l,j}+0} & \mathbf{T}_{t_{l,j}+1} & \cdots & \mathbf{T}_{t_{l,j}+m-1} \\ \mathbf{T}_{t_{l,j}+m} & \mathbf{T}_{t_{l,j}+m+1} & \cdots & \mathbf{T}_{t_{l,j}+2m-1} \\ \vdots & \vdots & \ddots & \vdots \\ \mathbf{T}_{t_{l,j}+(n-1)m} & \mathbf{T}_{t_{l,j}+(n-1)m+1} & \cdots & \mathbf{T}_{t_{l,j}+mn-1} \end{bmatrix} \tag{3}$$

Here, each $\mathbf{T}_{t_{l,j}+k}$ has a resolution of $H \times W$, making the resolution of $\mathbf{G}_{l,j}$ $mH \times nW$. Subsequently, bilinear interpolation is applied to resize $\mathbf{G}_{l,j}$ back to the original resolution $H \times W$, resulting in the final segment-end token grid $\text{Interp}(\mathbf{G}_{l,j}) \in \mathbb{R}^{H \times W \times d}$. The last frame of the segment is then updated with the generated summary token:

$$\mathbf{T}_{t_{l,j}+K_l-1} \xleftarrow{\text{update}} \text{Interp}(\mathbf{G}_{l,j}) \tag{4}$$

By processing all segments across layers, the token sequence is updated to incorporate multi-scale temporal dynamics, providing a richer representation for downstream tasks. The Pyramid Temporal Gridding module not only captures fine-grained and coarse-grained temporal interactions but also ensures computational efficiency, making it a robust foundation for spatiotemporal modeling in video understanding. The updated token sequence is subsequently passed to the Norm-based Spatial Pooling module for further refinement.

## 3.4 Norm-based Spatial Pooling

Video LLMs, such as LLaVA-Video, often rely on uniform 2D pooling or bilinear interpolation to downsample visual tokens in the spatial dimension. These methods, however, treat all spatial tokens equally, disregarding the inherent heterogeneity in their information content. As illustrated in Figure 3a, a typical image frame is dominated by background regions, while semantically salient objects occupy only a small fraction of the spatial grid. By applying equal-weighted downsampling, these critical regions (rich in visual information) are inadequately prioritized, resulting in substantial information loss and token redundancy. This limitation underscores the need for a more refined approach to spatial pooling that dynamically weighs tokens based on their semantic importance.

To address the aforementioned challenges, we design a novel spatial pooling method that dynamically prioritizes regions based on their information saliency while preserving critical visual features.

To identify an effective indicator of regional saliency, we conduct experiments on the validation set of the HKU-IS salient object detection dataset (Li & Yu, 2016). We perform both qualitative and quantitative analyses to examine the discrepancy in token norms between salient object areas and background regions, as illustrated in Figure 3. In Figure 3a, we visualize the area of the top 50% visual tokens in terms of L2 norm alongside the salient object ground truth. The results clearly demonstrate that regions corresponding to salient objects consistently exhibit high token norms, while redundant background regions are associated with low token norms. Furthermore, in Figure 3b, we plot the norm distributions of visual tokens for both background and salient object regions. The quantitative analysis confirms a significant discrepancy in token norms between these two regions. This insight helps us to establish token norm as a reliable metric for assessing regional information saliency.

Inspired by these findings, we introduce Norm-based Spatial Pooling (NSP), a dynamic pooling mechanism that leverages visual token norms to assign adaptive weights to each spatial location during the pooling process. By computing these weights based on the L2 norm of visual tokens, NSP selectively amplifies the representation of high-importance regions, such as salient objects, while diminishing the influence of low-importance backgrounds. This weighting strategy ensures that semantically rich visual information is prioritized and preserved, leading to more efficient and effective token representations. Through this adaptive pooling approach, NSP significantly enhances Video LLMs to focus on critical regions, ultimately improving the quality of visual token representations for downstream tasks.

Specifically, the input to NSP is the visual token sequence $\mathbf{T}_1, \mathbf{T}_2, \cdots, \mathbf{T}_N \in \mathbb{R}^{H \times W \times d}$, derived from the PTG module. Here, $N$ represents the number of frames, while $H$, $W$, and $d$ denote the height, width, and feature dimension of each token, respectively. The pooling operation employs a kernel size of $(k_H, k_W)$ and a stride of $(s_H, s_W)$. For an input visual token $\mathbf{T}_i$, let $\mathbf{t}$ denote the sliding window corresponding to the $h$-th row and $w$-th column of the output pooled token $\mathbf{T}_i^{\downarrow}(h, w)$. The elements of the sliding window are defined as:

$$\mathbf{t}_{m,n} = \mathbf{T}_i(h \cdot s_H + m, w \cdot s_W + n), \tag{5}$$

where $0 \leq m < k_H$, $0 \leq n < k_W$, and $h$, $w$ represent the spatial indices of the output feature map $\mathbf{T}_i^{\downarrow}$. For example, in a conventional 2x2 kernel, $m = 0$, $n = 0$ represents the token at the upper-left corner of the kernel window, while $m = 1$, $n = 1$ represents the token at the lower-right corner. For each visual feature $\mathbf{t}_{m,n}$ within the sliding window, we first calculate its $L_p$ norm, denoted as $\|\mathbf{t}_{m,n}\|_p$. This norm is then normalized into a weight $\alpha_{m,n}$ using the softmax function, ensuring that the weights sum to one across the window:

$$\alpha_{m,n} = \frac{\exp(\beta \|\mathbf{t}_{m,n}\|_p)}{\sum_{i=0}^{k_H-1} \sum_{j=0}^{k_W-1} \exp(\beta \|\mathbf{t}_{i,j}\|_p)}, \tag{6}$$

where $\beta$ is a temperature parameter that controls the sharpness of the weight distribution. Finally, the pooling result for each sliding window is obtained as a weighted summation of the visual features:

$$\mathbf{T}_i^{\downarrow}(h, w) = \sum_{m=0}^{k_H-1} \sum_{n=0}^{k_W-1} \alpha_{m,n} \cdot \mathbf{t}_{m,n}. \tag{7}$$

During the pooling process, the NSP mechanism leverages the intrinsic relationship between visual token norms and semantic saliency to dynamically prioritize high-importance regions. By integrating a softmax-based weighting scheme, NSP adaptively enhances the representation of semantically rich visual features while suppressing less informative background regions. This approach not only preserves critical visual details but also maintains computational efficiency, making it a plug-and-play enhancement for existing Video LLMs. As a result, NSP significantly improves the quality of visual token representations, enabling more robust and accurate video understanding.

## 4 EXPERIMENTS

### 4.1 EXPERIMENTAL SETTINGS

**Baselines** To validate the effectiveness of the proposed method, we employ two LLaVA family models as backbones: LLaVA-OneVision-7B (Li et al., 2024a) and LLaVA-Video-7B (Zhang et al.,

| Model | Long Video Understanding | | | General Video Understanding | | |
|---|---|---|---|---|---|---|
| | VideoMME | LongV.Bench | EgoSchema | NexT-QA | TempCompass | MVBench |
| VideoLLaMA2.1-7B | 54.9 | - | 53.1 | - | 56.8 | 57.3 |
| LongVA-7B | 52.6 | - | - | 68.3 | - | - |
| IXC-2.5-7B | 55.8 | - | - | - | - | 69.1 |
| InternVideo2-7B | - | - | 60.0 | - | - | 67.2 |
| Oryx-1.5-7B | 58.8 | 56.3 | - | 81.8 | - | 67.6 |
| NVILA-8B | 64.2 | 57.7 | - | 82.2 | 69.7 | 68.1 |
| mPLUG-Owl3-8B | 53.5 | 52.1 | - | 78.6 | - | 54.5 |
| Apollo-7B | 61.3 | 58.5 | - | - | 64.9 | - |
| LLaVA-OneVision-7B | 58.2 | 56.5 | 60.1 | 79.4 | 64.2 | 56.7 |
| + Ours | 59.0 | 56.7 | 62.1 | 79.6 | 64.4 | 58.0 |
| | (+0.8%) | (+0.2%) | (+2.0%) | (+0.2%) | (+0.2%) | (+1.3%) |
| LLaVA-Video-7B | 63.3 | 58.2 | 57.3 | 83.2 | 65.4 | 58.6 |
| + Ours | 64.2 | 60.1 | 57.8 | 83.8 | 66.1 | 59.8 |
| | (+0.9%) | (+1.9%) | (+0.5%) | (+0.6%) | (+0.7%) | (+1.2%) |

Table 1: Overall comparison with state-of-the-art methods on long-form and general video understanding benchmarks (%). The best performance among all methods is underlined.

2024d). These models serve as the foundation for our evaluation, enabling us to assess the performance improvements brought by our approach in video understanding tasks. Additionally, we compare against a diverse set of baseline models to ensure a comprehensive analysis. These include VideoLLaMA2.1-7B (Cheng et al., 2024), LongVA-7B (Zhang et al., 2024c), IXC-2.5-7B (Zhang et al., 2024b), InternVideo2-7B (Wang et al., 2024), Oryx-1.5-7B (Liu et al., 2024e), NVILA-8B (Liu et al., 2024d), mPLUG-Owl3-8B (Ye et al., 2024), and Apollo-7B (Zohar et al., 2024). These models represent state-of-the-art advancements in video and multi-modal understanding, providing a robust framework for benchmarking our method.

**Benchmarks** To comprehensively evaluate the proposed method, we conduct experiments on benchmarks from both Long Video Understanding and General Video Understanding domains. For **Long Video Understanding**, we adopt VideoMME (Fu et al., 2024), LongVideoBench (Wu et al., 2025), and EgoSchema (Mangalam et al., 2024), which focus on capturing temporal dependencies and reasoning over extended video sequences. These benchmarks are specifically designed to assess the ability of models to process and interpret long-range visual information. For **General Video Understanding**, we utilize NexT-QA (Xiao et al., 2021), TempCompass (Liu et al., 2024c), and MVBench (Li et al., 2024b). These datasets are widely recognized for their diverse scenarios and challenging complexity.

## 4.2 IMPLEMENTATION DETAILS

Our experiments are conducted on NVIDIA L20 GPUs with Ubuntu 22.04. We utilize the *lmms-eval* (Zhang et al., 2024a) library for model modification and evaluation. To maintain consistency in computational load, we ensure identical input frame counts for the visual encoder and token counts for the LLM with the original models. For the LLaVA-OneVision model, we follow the original setup to feed 32 frames into the visual encoder, while for the LLaVA-Video model, we adhere to the original setting of 64 input frames. We employ a spatial pooling strategy with a kernel size and stride of 2. As for hyperparameter, we set the temperature parameter $\beta = 1$ and the norm order $p = 2$, optimizing for stable performance across diverse video understanding tasks. This setup allowed us to validate the efficacy of our proposed methods while maintaining computational fairness in comparative evaluations.

| Method | VideoMME | L.V.Bench | EgoSchema |
|---|---|---|---|
| **Upper Bound (Full Tokens)** | | | |
| LLaVA-Video | 63.3 | 58.2 | 57.3 |
| **Token Budget Ratio: 30%** | | | |
| FastV | 59.3 | 53.5 | 51.3 |
| PruMerge | 59.9 | 54.7 | 50.9 |
| FasterVLM | 60.1 | 55.8 | 52.6 |
| VisionZip | 58.3 | 53.2 | 53.0 |
| FrameFusion | 61.3 | 56.0 | 53.0 |
| **Ours** | **62.0** | **58.1** | **56.0** |
| **Token Budget Ratio: 50%** | | | |
| FastV | 62.2 | 55.7 | 54.7 |
| PruMerge | 61.3 | 56.9 | 54.6 |
| FasterVLM | 61.7 | 56.4 | 56.2 |
| VisionZip | 60.6 | 56.8 | 54.2 |
| FrameFusion | **62.6** | 57.6 | 55.8 |
| **Ours** | 62.5 | **58.9** | **57.1** |

Table 2: Comparison with token reduction baselines on LLaVA-Video-7B (%).

## 4.3 COMPARISON STUDY

**Comparison with SOTA**  We first present the overall results across all datasets in Table 1, which reveals the following key observations: (1) When integrated with the original LLaVA-OneVision and LLaVA-Video models, our proposed method achieves consistent performance improvements across all evaluated datasets. (2) The ST-GridPool mechanism significantly enhances long-term video understanding, enabling our method to surpass existing 7B models on long-form benchmarks like VideoMME, LongVideoBench, and Egoschema (3) Our method consistently demonstrates competitive or leading performance in general video understanding tasks. For example, LLaVA-Video achieves 66.1% on TempCompass, making it closer to SOTA performance of NVILA-8B. These advancements highlight the adaptability of our approach across diverse video understanding scenarios while maintaining the training-free advantage and efficient computational load.

**Comparison with Token Reduction Methods**  To further evaluate the effectiveness of our proposed method as an efficient visual token compression strategy, we compare it directly with several leading token reduction methods. As shown in Table 4, the experiments are performed on the LLaVA-Video-7B model using two different token budgets: 50% and 30%. The performance of each method is evaluated on three long-video understanding benchmarks. (1) With the 50% token budget, our method shows highly competitive performance. It achieves the highest scores on the L.V.Bench and EgoSchema datasets, and its performance on VideoMME is comparable to the best-performing method, FrameFusion. (2) The advantages of our method become more evident under the stricter 30% token budget, which involves a higher compression rate. Under these conditions, our method achieves the best performance on all three benchmarks, outperforming all other methods. (3) In summary, our proposed method performs well in standard compression scenarios and is also highly robust under high-compression (low-budget) conditions. This indicates that our approach can more effectively identify and preserve the key information crucial for video understanding, especially when the token budget is highly limited.

## 4.4 ABLATION STUDY

**Effectiveness of Different Components**  To validate the effectiveness of individual components in our method, we conduct ablation studies on two key components: Norm-based Spatial Pooling (NSP) and Pyramid Temporal Gridding (PTG). As shown in Table 3, three key observations emerge: (1) The full integration of NSP and PTG achieves optimal performance, showing consistent

| Model | VideoMME | LongV.Bench | MVBench |
|---|---|---|---|
| Baseline | 63.3 | 58.2 | 58.6 |
| Ours w/o NSP | 63.8 | 59.2 | 59.1 |
| Ours w/o PTG | 63.6 | 59.8 | 58.8 |
| **Ours** | **64.2** | **60.1** | **59.8** |

Table 3: Ablation results of different components (%).

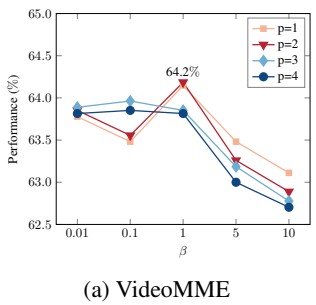
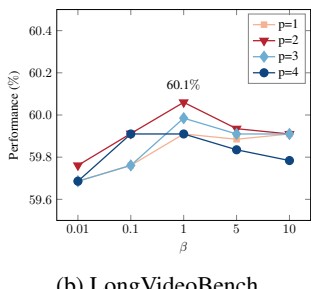

(a) VideoMME                (b) LongVideoBench

Figure 4: Ablation study results for different values of temperature $\beta$ and norm order $p$.

gains over both standalone components. This demonstrates their complementary roles in optimizing spatiotemporal token downsampling for video understanding. (2) Removing PTG (Ours w/o PTG) results in a performance drop, demonstrating its role in improving temporal feature aggregation. The absence of NSP (Ours w/o NSP) also leads to reduced performance, highlighting its contribution to optimizing spatial feature representation. (3) While baseline modifications with single components improve performance, their isolated effects remain suboptimal. Our unified framework leverages NSP for spatial saliency alignment and PTG for multiscale temporal reasoning, establishing their joint necessity for achieving state-of-the-art results.

**Impact of $\beta$ and $L_p$** We analyze two critical hyperparameters in the Norm-aware Spatial Pooling (NSP) module: the temperature coefficient $\beta$ and the norm order $p$ of $L_p$. To evaluate their impact, we conduct experiments on the LLaVA-Video backbone with varying $\beta = \{0.01, 0.1, 1, 5, 10\}$ and $p = \{1, 2, 3\}$, and summarize results on VideoMME and LongVideoBench in Figure 4. Some observations can be made as follows: (1) Performance first rises and then declines with increasing $\beta$: Optimal results are achieved at $\beta = 1$, while extreme values (e.g., $\beta = 5$ or $\beta = 10$) degrade accuracy, likely due to over-smoothing or unstable feature activation. (2) Increasing $p$ beyond 2 leads to gradual performance degradation. The $L_2$-norm achieves the highest scores across both datasets, balancing spatial sparsity and feature discriminability. (3) Despite sensitivity to extreme $\beta$ or $p$ values, our method shows consistent trends across different datasets. Variations within experiments yield subtle performance fluctuations, highlighting its robustness. The interplay between $\beta$ and $L_p$ reveals a delicate balance: $\beta = 1$ ensures appropriate sharpness in activation distributions, while $L_2$-norm optimally aggregates spatial features without over-sparsity.

## 4.5 COMPUTATIONAL COST ANALYSIS

We analyzed the computational efficiency of ST-GridPool under a strict 30% token budget. The experiments compare our method against the baseline LLaVA-Video-7B model in terms of inference time and peak GPU memory usage when processing different numbers of input frames. As shown in Fig. 5, our method achieves a dual optimization in computational efficiency. For inference time (left figure), ST-GridPool demonstrates a significant advantage, and the savings become more pronounced as the number of input frames increases. For GPU memory usage (right figure), our method also outperforms the baseline, consistently saving peak memory. In summary, ST-GridPool not only enhances video understanding performance but also substantially reduces both inference time and memory usage, proving its value as a highly efficient enhancement solution.

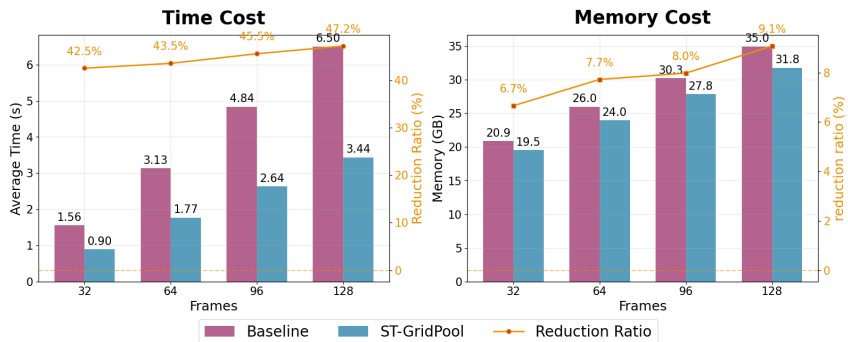

Figure 5: Computational cost comparison between our method and the baseline LLaVA-Video model in inference time and GPU memory usage under a 30% token budget.

## 4.6 QUALITATIVE ANALYSIS OF EXAMPLES

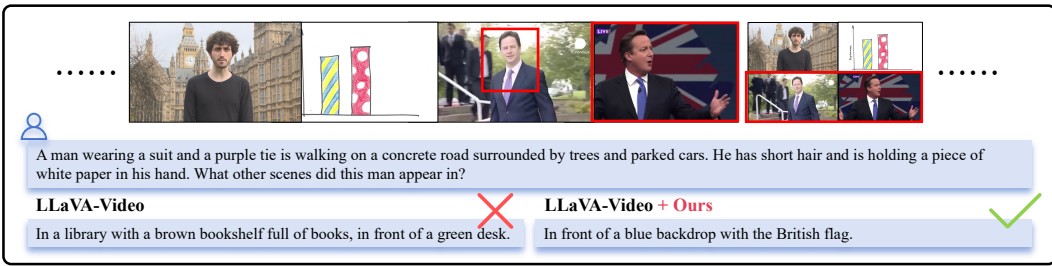

Figure 6: Response examples of LLaVA-Video with and w/o Ours from LongVideoBench dataset.

We conduct a qualitative analysis using diverse samples from the LongVideoBench dataset to evaluate the performance of our method in fine-grained video understanding tasks. As illustrated in Figure 6, when processing long videos, our method effectively captures and correlates distant spatio-temporal information, enabling a holistic understanding of extended events. These observations highlight our method's ability to handle both short-term spatial reasoning and long-term temporal reasoning. By integrating multi-scale temporal gridding and adaptive spatial pooling, our approach achieves robustness and precision in complex video understanding scenarios. Such capabilities are crucial for applications requiring detailed analysis of dynamic and prolonged activities.

## 5 CONCLUSION

In this paper, we presented ST-GridPool, a training-free visual token enhancement method designed to bridge the gap between efficient token compression and the preservation of intricate spatiotemporal interactions in Video LLMs. Our framework integrates Pyramid Temporal Gridding (PTG) to capture multi-grained temporal dynamics through hierarchical segments and Norm-based Spatial Pooling (NSP), which prioritizes high-information regions by leveraging the positive correlation between token norms and semantic richness. Extensive experiments on multiple benchmarks demonstrate that ST-GridPool significantly improves the performance of models like LLaVA-Video without requiring retraining, showing particular robustness under strict token budgets. Furthermore, our approach achieves a dual optimization by substantially reducing both inference latency and peak GPU memory usage as input frame counts increase. As a scalable and plug-and-play solution, ST-GridPool offers a powerful paradigm for enhancing visual token representations without the need for costly architectural modifications.

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

# A   COMPARISON WITH TOKEN REDUCTION METHODS

| Method | VideoMME | L.V.Bench | EgoSchema |
|---|---|---|---|
| **Upper Bound (Full Tokens)** | | | |
| NVILA | 61.5 | 56.3 | 52.9 |
| **Token Reduction Ratio: 30%** | | | |
| FastV | 57.9 | 53.0 | 49.7 |
| PruMerge | 58.2 | 53.4 | 47.5 |
| FasterVLM | 60.1 | 53.0 | 49.3 |
| VisionZip | 59.1 | 50.9 | 48.9 |
| FrameFusion | 58.8 | **54.9** | 51.3 |
| **Ours** | **59.9** | 54.6 | **52.0** |
| **Token Reduction Ratio: 50%** | | | |
| FastV | 58.9 | 53.9 | 50.2 |
| PruMerge | 57.6 | 53.9 | 48.9 |
| FasterVLM | 60.8 | 53.0 | 50.5 |
| VisionZip | 60.5 | 54.4 | 50.3 |
| FrameFusion | 59.4 | 54.8 | 52.6 |
| **Ours** | **61.4** | **55.6** | **53.1** |

Table 4: Comparison with token reduction baselines on NVILA-Video-8B with 64 input frames (%).

To validate our token reduction method, we conducted experiments on the NVILA-Video-8B model with token reduction ratios of 30% and 50%, evaluating performance on three long-video benchmarks (VideoMME, LongVideoBench, and EgoSchema). To ensure a fair comparison of computational efficiency and resource usage with LLaVA-Video-7B, we establish a consistent setup by also setting the input frame count for NVILA to 64. The results in Table 4 demonstrate our method's superior performance. At a 30% reduction, our approach is highly competitive and achieves the top score on EgoSchema. Its advantage becomes even more pronounced at a 50% reduction, where our method ranks first across all three benchmarks. Notably, at this high compression rate, our method's performance not only comes remarkably close to the full-token upper bound but even surpasses it on the EgoSchema benchmark. This strongly validates the effectiveness and generalizability of our approach, proving it can significantly reduce computational cost while maintaining, and in some cases even exceeding, the performance of the original full-token model.

# B   COMPARISON WITH ALTERNATIVES

We further compare our method with training-free alternative approaches, including IG-VLM (Kim et al., 2024), SF-LLaVA (Xu et al., 2024b), and TS-LLaVA (Qu et al., 2024). While these methods have shown promising results on image-language models like LLaVA-Next (Liu et al., 2024a), we adapt them to LLaVA-Video model under fair experimental settings: adjusting parameters to maintain identical input frames and token counts as the original models. As shown in Table 5, our analysis reveals two key findings: (1) Directly transplanting these image-focused methods to video-language models yields unsatisfactory outcomes. SF-LLaVA and TS-LLaVA achieve limited improvements on specific datasets, but their overall performance remains unstable and non-substantive. Notably, applying IG-VLM alone leads to significant performance degradation. (2) Our method achieves the most substantial gains across all metrics, outperforming all alternatives with improvements on VideoMME, LongVideoBench, and MVBench respectively compared to the original baseline. This demonstrates the unique effectiveness of our approach in aligning visual-language reasoning patterns for video understanding tasks.

| Model | VideoMME | LongV.Bench | MVBench |
|---|---|---|---|
| Baseline | 63.3 | 58.2 | 58.6 |
| + IG-VLM | 60.6 | 55.9 | 54.2 |
| + SF-LLaVA | 63.4 | 58.9 | 58.9 |
| + TS-LLaVA | 63.8 | 59.6 | 57.1 |
| **+ Ours** | **64.2** | **60.1** | **59.8** |

Table 5: Comparison with training-free alternative methods on the LLaVA-Video-7B baseline (%).

| Method | VideoMME | LongV.Bench | MVBench | EgoSchema |
|---|---|---|---|---|
| Image-gridding | 58.3 | 56.3 | 56.5 | 56.6 |
| **Token-gridding (Ours)** | **59.0** | **56.7** | **58.0** | **62.1** |

Table 6: Comparison with image-gridding and token-gridding (ours) on the LLaVA-OneVision-7B (%).

## C  ABLATION STUDY ON IMAGE-GRIDDING VS TOKEN-GRIDDING

In Pyramid Temporal Gridding (PTG), we introduce token-level gridding, which diverges from the image-level gridding employed by previous methods such as IG-VLM (Kim et al., 2024) and TS-LLaVA (Qu et al., 2024). To examine the performance differences between these approaches, we conducted experiments using image-gridding, where the PTG module processes information at the image level, akin to IG-VLM. In contrast, our method applies token-gridding on token representations. Results are shown in table 6 and table 7, which demonstrate that the token-grid strategy consistently outperforms the image-grid approach across both LLaVA-OneVision-7B and LLaVA-Video-7B configurations. This superiority highlights the advantages of fine-grained token-level processing, which preserves richer spatio-temporal features and enables more precise modeling of video dynamics. These findings validate the effectiveness of our token-gridding strategy, showcasing its ability to enhance performance while maintaining computational efficiency.

## D  IMPACT OF TEMPORAL GRIDDING CONFIGURATION

We investigate the impact of temporal pyramid configurations in the Pyramid Temporal Gridding (PTG) module, focusing on two critical design choices: the number of pyramid layers ($L$) and the segment length at each layer. In experiments, each layer's segment length doubles the previous one (e.g., $(8, 16, 32)$). Results on VideoMME and LongVideoBench, shown in Figure 7, reveal the following obsevations: (1) For a fixed $L$, performance initially improves with increasing maximum segment length but declines after a threshold, suggesting overly long segments may dilute fine-grained motion patterns. Medium-length segments balance local detail capture and global temporal context, whereas excessively long segments introduce noise from irrelevant temporal regions. (2) Increasing $L$ systematically boosts performance, with the best global results achieved at $L = 3$ using $(8, 16, 32)$ segments. As $L$ grows from 1 to 3, the ideal maximum segment length increases from 8 ($L = 1$) to 32 ($L = 3$). Deeper pyramids enable hierarchical modeling of multi-scale temporal dependencies, leveraging shorter segments for local actions and longer ones for macro-event integration.

## E  IMPACT OF SPATIAL POOLING SHAPE

We explore the impact of spatial pooling shape (i.e., kernel size) in the Norm-aware Spatial Pooling (NSP) module. Experiments are conducted on the LLaVA-Video backbone with kernel sizes spanning $(1, 1), (2, 2), \ldots, (6, 6)$, and results on VideoMME and LongVideoBench are depicted in Figure 8. It is observed that both datasets achieve peak performance at a kernel size of $(2, 2)$. Smaller kernels $(1, 1)$ yield suboptimal results, as overly localized receptive fields fail to capture contextual spatial relationships, limiting feature aggregation. Performance declines progressively for kernels

| Method | VideoMME | LongV.Bench | MVBench | EgoSchema |
|---|---|---|---|---|
| Image-gridding | 62.8 | 59.0 | 58.4 | 57.1 |
| **Token-gridding (Ours)** | **64.2** | **60.1** | **59.8** | **57.8** |

Table 7: Comparison with image-grid and token-grid (ours) on the LLaVA-Video-7B (%).

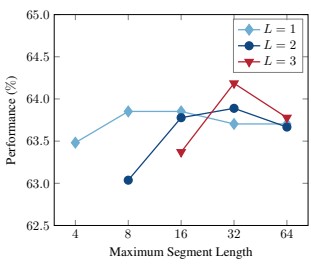

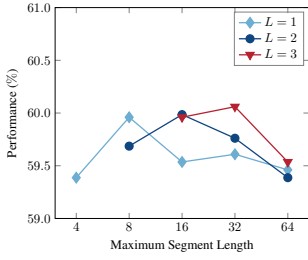

(a) VideoMME

(b) LongVideoBench

Figure 7: Ablation study results for different values of level $L$ and maximum segment length on VideoMME and LongVideoBench.

larger than $(2, 2)$, with $(6, 6)$ delivering the lowest scores. This degradation may be caused by the oversmoothing effect and local structural misalignment of broad receptive fields.

## F  FINE-GRAINED MULTI-TASK ANALYSIS ON MVBENCH

| Model | AA | AC | AL | AP | AS | CO | CI | EN | ER | FA | FP | MA | MC | MD | OE | OI | OS | ST | SC | UA | Avg. |
|---|---|---|---|---|---|---|---|---|---|---|---|---|---|---|---|---|---|---|---|---|---|
| GPT-4V | 72.0 | 39.0 | 40.5 | 63.5 | 55.5 | 52.0 | 11.0 | 31.0 | 59.0 | 46.5 | 47.5 | 22.5 | 12.0 | 12.0 | 18.5 | 59.0 | 29.5 | 83.5 | 45.0 | 73.5 | 43.5 |
| Video-ChatGPT (Maaz et al., 2023) | 62.0 | 30.5 | 20.0 | 26.0 | 23.5 | 33.0 | 35.5 | 29.5 | 26.0 | 22.5 | 29.0 | 39.5 | 25.5 | 23.0 | 54.0 | 28.0 | 40.0 | 31.0 | 48.5 | 26.5 | 32.7 |
| Video-LLaMA (Zhang et al., 2023) | 51.0 | 34.0 | 22.5 | 25.5 | 27.5 | 40.0 | 37.0 | 30.0 | 21.0 | 29.0 | 32.5 | 32.5 | 22.5 | 22.5 | 48.0 | 40.5 | 38.0 | 43.0 | 45.5 | 39.0 | 34.1 |
| VideoChat (Li et al., 2023b) | 56.0 | 35.0 | 27.0 | 26.5 | 33.5 | 41.0 | 36.0 | 23.5 | 23.5 | 33.5 | 26.5 | 42.5 | 20.5 | 25.5 | 53.0 | 40.5 | 30.0 | 48.5 | 46.0 | 40.5 | 35.5 |
| PLLaVA (Xu et al., 2024a) | 55.5 | 39.5 | 26.0 | 49.0 | 58.0 | 53.5 | 31.0 | 30.5 | 48.0 | 41.0 | 42.0 | 52.0 | 42.0 | 23.5 | 56.0 | 61.0 | 36.0 | 82.0 | 45.0 | 61.0 | 46.6 |
| ST-LLM (Liu et al., 2024b) | 84.0 | 36.5 | 31.0 | 53.5 | 66.0 | 46.5 | 58.5 | 34.5 | 41.5 | 44.0 | 44.5 | 78.5 | 56.5 | 42.5 | 80.5 | 73.5 | 38.5 | 86.5 | 43.0 | 58.5 | 54.9 |
| VideoChat2 (Li et al., 2024b) | 83.5 | 37.0 | 44.0 | 58.0 | 75.5 | 47.0 | 72.5 | 35.0 | 37.0 | 50.5 | 66.5 | 87.5 | 64.5 | 47.5 | 87.5 | 74.5 | 45.0 | 82.5 | 51.0 | 60.5 | 60.4 |
| **LLaVA-OV + Ours** | 68.0 | 48.0 | 55.0 | 57.5 | 71.0 | 71.0 | 47.0 | 35.0 | 52.0 | 48.0 | 54.0 | 71.5 | 47.0 | 31.5 | 57.5 | 82.5 | 35.5 | 94.5 | 51.5 | 80.0 | 58.0 |
| **LLaVA-Video + Ours** | 66.0 | 53.0 | 58.5 | 59.5 | 72.5 | 77.0 | 51.0 | 28.5 | 54.5 | 49.0 | 55.5 | 70.0 | 45.5 | 39.0 | 58.5 | 84.5 | 40.0 | 91.5 | 60.0 | 81.5 | 59.8 |

Table 8: Multi-task analysis results on MVBench (%). The best performance among all methods is underlined.

To further analyze the fine-grained video understanding capabilities of our method, we present the detailed performance of compared methods across sub-tasks on the MVBench dataset, as shown in table 8. Based on the results, the following observations can be made: (1) Our method achieves comparable performance to the state-of-the-art baseline models, demonstrating its robustness across a wide range of tasks. (2) Our method substantially outperforms baselines in tasks requiring spatial localization and temporal granularity, such as Action Localization (AL), Unexpected Action (UA), Scene Transition (ST), State Change (SC). These improvements highlight the effectiveness of our proposed Norm-based Spatial Pooling and Pyramid Temporal Gridding in enhancing spatial precision and temporal granularity. (3) Despite the advancements, our method lags behind SOTA in tasks requiring long-horizon reasoning or fine-grained motion analysis, highlighting opportunities to better integrate cross-frame dependencies.

## G  EXTENSIVE QUALITATIVE ANALYSIS OF EXAMPLES

We illustrate some additional examples from LongVideoBench dataset in fig. 9. Visual comparisons demonstrate that our method (LLaVA-Video-7B + Ours) significantly outperforms the baseline in capturing fine-grained spatiotemporal details. One key limitation of baseline models is their inability to resolve temporal dependencies, even when all event components are clearly present in the video. For instance, in the first example, when asked about event sequencing, standard Video LLMs misorder actions, confusing co-occurrence with causality. Our temporal gridding strategy tackles this issue by explicitly organizing spatiotemporal tokens into chronological lattices, allowing the

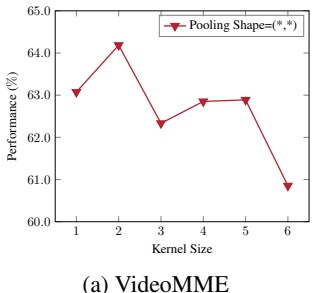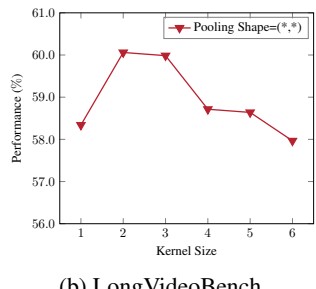

(a) VideoMME                    (b) LongVideoBench

Figure 8: Ablation study results for different pooling shape on VideoMME and LongVideoBench.

model to accurately disentangle sub-frame temporal relationships. The integration of norm-based recalibration enhances the model's ability to localize subtle visual cues. In the second example, the baseline incorrectly identifies a clothing pattern as "solid blue lines" due to its failure to detect the faint dashed stitching. In contrast, our method successfully recognizes the discontinuous texture through normalized feature recalibration. These examples collectively highlight how our method combines spatial refinement and temporal coherence to achieve narratively consistent and visually precise video reasoning.

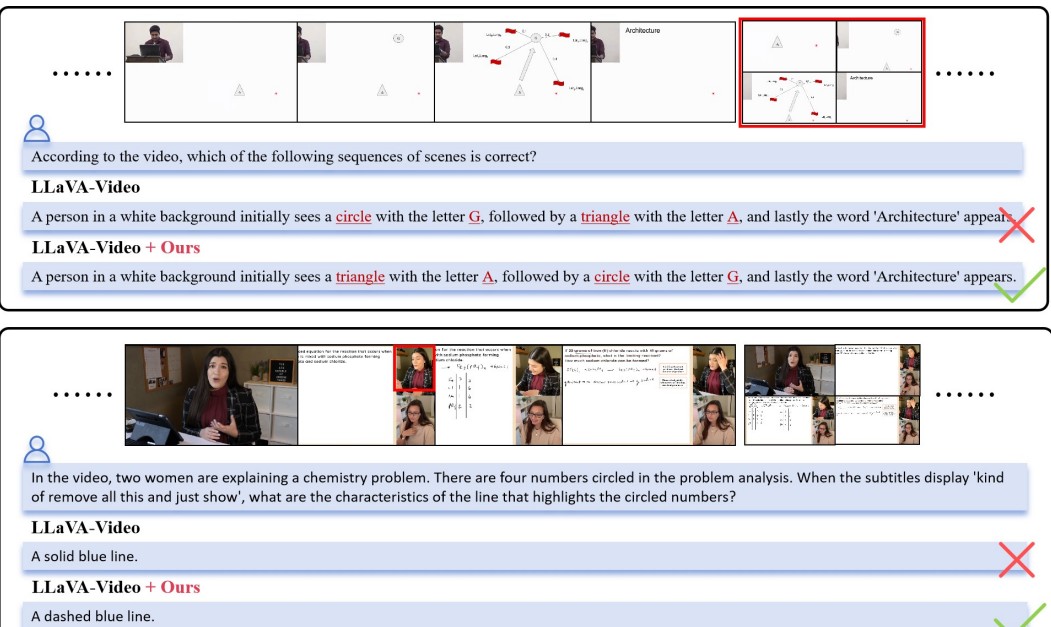

Figure 9: The other output examples of LLaVA-Video model without and with our method on LongVideoBench dataset.

