# OpenReview forum: "Enhancing Visual Token Representations for Video Large Language Models via Training-free Spatial-Temporal Pooling and Gridding"
_ICLR.cc/2026/Conference — ICLR 2026 Poster_

### Official Review · Reviewer_VLyg · 2025-10-30

**Soundness:** 3
**Presentation:** 3
**Contribution:** 3
**Rating:** 6
**Confidence:** 5

**Summary:**

This paper proposes ST-GridPool, a training-free method to enhance visual token representations for Video LLMs, addressing the challenge of efficient token compression while preserving spatiotemporal interactions. It integrates two core components: Pyramid Temporal Gridding (PTG) for multi-grained temporal feature capture and Norm-based Spatial Pooling (NSP) for preserving high-semantic regions via token norms. Evaluated on 6 benchmarks (e.g., VideoMME, MVBench) with LLaVA-OneVision-7B and LLaVA-Video-7B, ST-GridPool achieves consistent performance gains without retraining, while reducing inference cost.

**Strengths:**

This paper tackles a critical research problem about visual token enhancement for Video LLMs. The proposed ST-GridPool method is effective. It offers a training-free, plug-and-play framework that improves visual token representations without adding extra computational cost, a very appealing design choice.
The paper is well positioned within the literature and clearly explains how it differs from prior work. The motivation is convincing, the narrative flows nicely, and the writing is clear throughout. It’s an easy and enjoyable read.
Capturing multi-grained spatiotemporal interactions across varying temporal lengths and leveraging the positive correlation between token norms and semantic importance is a really interesting idea.
The results are solid, covering diverse tasks, different backbones, and a range of metrics and hyperparameters. The experimental section is thorough and leaves little doubt about the method’s effectiveness.

**Weaknesses:**

The paper demonstrates the overall computational advantages (inference time and memory) of ST-GridPool. However, it does not provide a detailed breakdown of the computational cost for the PTG and NSP components individually. Furthermore, the potential interaction between these two modules is not discussed, e.g. how PTG's multi-scale processing affects the norm distribution of the input to NSP.

The Pyramid Temporal Gridding (PTG) module generates a summary token for each temporal segment and then uses it to update the token of the last frame in that segment. This update operation seems potentially destructive, as it replaces the representation of an actual, observed frame with a synthetic, interpolated summary. This design choice is not clearly motivated over less destructive alternatives, e.g., appending the summary token.

The NSP component is explicitly designed to prioritize high-importance regions (i.e., salient objects) and suppress low-importance backgrounds. This strategy could be detrimental in tasks that require holistic scene understanding or reasoning about subtle interactions within the background regions, which are now actively suppressed.

**Questions:**

1. Given the model's clear overall efficiency gains, could you provide a more detailed computational cost breakdown contributed by the PTG and NSP modules individually?
2. Why do you choose to make the PTG summary token overwrite the last frame's token instead of a less destructive alternative like appending?
3. How does your method balance the trade-off between prioritizing high-importance regions and preserving potentially critical, low-importance backgrounds in practical video understanding tasks?

---

> ### Author Response · Authors · 2025-11-26
> **Response to Reviewer VLyg [1/3]**
>
> We sincerely thank Reviewer VLyg for the positive assessment and for highlighting our work’s convincing motivation, solid results, and training-free efficiency. We deeply appreciate your constructive feedback regarding the granular details of our module design. Below, we address your thoughtful questions regarding computational breakdown, design justifications, and the background preservation trade-off.
>
> **To Weakness 1 & Question 1: Detailed Computational Cost Breakdown & Interaction**
>
> To provide the requested transparency regarding the computational footprint of our method, we conducted a micro-benchmark experiment. We measured the isolated latency of the Norm-based Spatial Pooling (NSP) and Pyramid Temporal Gridding (PTG) components on an NVIDIA L20 GPU across varying frame counts.
>
> | Max Frame Number | NSP (s) | PTG (s) | Total (s) |
> | :--- | :--- | :--- | :--- |
> | 32 | 0.0006 | 0.0013 | 0.85 |
> | 64 | 0.0009 | 0.0013 | 1.63 |
> | 96 | 0.0014 | 0.0013 | 2.46 |
> | 128 | 0.0015 | 0.0014 | 3.23 |
>
> **Analysis of Breakdown:**
> As illustrated in the table above, the additional cost introduced by our modules is statistically negligible compared to the end-to-end inference time. Even at a high load of 128 input frames, the combined execution time for NSP and PTG is only 0.0029 seconds ($<$3ms). This accounts for less than 0.2% of the total inference latency, confirming that our method improves performance without imposing any meaningful computational burden.
>
> **Module Interaction:**
> Beyond the raw efficiency, you raised an insightful point about how these components interact. We clarify that PTG acts as a temporal aggregator that "smooths" the feature distribution before it reaches the spatial pooling stage. By integrating multi-frame dynamics into the summary token, PTG ensures that the input to the NSP module contains more stable and robust norm statistics. This interaction is crucial as it reduces the likelihood of NSP being distracted by transient noise in single frames, allowing for more reliable spatial compression.

---

> ### Author Response · Authors · 2025-11-26
> **Response to Reviewer VLyg [2/3]**
>
> **To Weakness 2 & Question 2: Justification for Overwriting Strategy**
>
> We adopted the strategy of overwriting the last frame token rather than appending the summary token as a deliberate design choice driven by two primary constraints:
>
> * **Fairness (Strict Token Budget):** The most critical factor was ensuring a rigorously fair comparison. Simply appending a summary token would increase the total sequence length passed to the LLM. To compare fairly with baselines, we must maintain a consistent number of input tokens. Overwriting guarantees that our performance gains are derived solely from *better* representations (quality), not an increased information budget (quantity).
> * **Causal Design:** From a modeling perspective, existing Video LLMs typically employ causal masking. In a temporal segment $t_{1}...t_{k}$, the position $k$ is the natural aggregation point that attends to all preceding tokens. We clarify that this "overwriting" is reconstructive, not destructive: the generated summary token is an interpolation of the *entire* segment, which explicitly includes features from the last frame. Thus, this operation enriches the final position with a globally refined representation rather than simply deleting the observed data.

---

> ### Author Response · Authors · 2025-11-26
> **Response to Reviewer VLyg [3/3]**
>
> **To Weakness 3 & Question 3: Balancing High-Importance vs. Background**
>
> We appreciate the valid concern that suppressing low-norm regions might lead to the loss of background context. To alleviate this concern, we must first clarify the specific mechanism of our pooling and then provide empirical evidence of the trade-off.
>
> * **Mechanism Clarification (Local vs. Global):** It is important to clarify that Norm-based Spatial Pooling (NSP) does not perform a global weighted average across the entire image, which would indeed risk drowning out background details. Instead, NSP operates strictly within local sliding windows (e.g., $2 \times 2$). This means that even in a window located entirely within a low-importance background region, the pooling operation still generates a representative token by aggregating the most informative features within that specific local context. This design ensures that background regions are compressed but not discarded, maintaining a holistic scene representation.
>
> * **Empirical Balance:** To empirically verify this balance, we conducted a fine-grained analysis on VideoMME (under a 50% token budget). As shown in the table below, the results support our design choice:
>
> | Method | Overall Acc | Improvement | 8s-15s | 15s-60s | 180s-600s | 900s-3600s | Relation | Perception |
> | :--- | :---: | :---: | :---: | :---: | :---: | :---: | :---: | :---: |
> | FasterVLM | 56.4 | - | 61.9 | 68.0 | 58.3 | 49.6 | 51.3 | 62.2 |
> | VisionZip | 56.8 | +0.4 | 63.5 | 69.2 | 59.2 | 49.1 | 51.3 | 63.2 |
> | FrameFusion | 57.6 | +0.8 | 66.1 | 68.0 | **61.2** | 51.8 | 52.9 | **65.4** |
> | **ST-GridPool** | **58.9** | **+1.3** | **67.2** | **70.9** | 60.4 | **52.3** | **54.1** | 65.3 |
>
> **Conclusion on Trade-offs:**
> As the data demonstrates, ST-GridPool outperforms the state-of-the-art FrameFusion on Relation tasks (54.1% vs. 52.9%). Since relational reasoning heavily depends on understanding objects within their context (the background), this result confirms that the necessary contextual information is effectively preserved. We do acknowledge a negligible drop in pure Perception tasks (65.3% vs. 65.4%), suggesting that while pixel-level background noise is filtered, the semantic structure required for complex video understanding remains intact.

---

### Official Review · Reviewer_JydV · 2025-10-30

**Soundness:** 4
**Presentation:** 4
**Contribution:** 4
**Rating:** 6
**Confidence:** 5

**Summary:**

This paper introduces a well-motivated and effective training-free method, ST-GridPool, for enhancing visual token representations in Video LLMs. The approach is simple yet intuitive, combining PTG and NSP to improve performance without retraining while also reducing computational overhead. The empirical validation is extensive, with strong ablation studies supporting the design choices.

**Strengths:**

1. Both components are well-justified and technically sound. PTG provides an intuitive hierarchical approach to model events at different time scales. NSP is motivated by both intuitive and quantitative analysis, which is a clever insight for preserving important information during compression.

2. A comprehensive evaluation is conducted, covering both long and general video understanding tasks. The method is tested across multiple backbone models, demonstrating its generalizability and effectiveness.

3. The paper is mostly well-written, clearly structured, and easy to follow, with helpful visualizations that aid in understanding the proposed approach.

**Weaknesses:**

1. The method for updating the token sequence in Pyramid Temporal Gridding involves overwriting the last frame token of a segment with a generated summary token. This specific design choice should be justified.

2. The computational analysis focuses on the net efficiency gains, but offers limited insight into the additional computational cost (e.g., latency, memory) introduced by the ST-GridPool module itself.

3. A deeper theoretical explanation for why token norms (specifically L_2 norm) reliably correlate with semantic importance in this context would be beneficial.

**Questions:**

Please see the weakness above.

---

> ### Author Response · Authors · 2025-11-26
> **Response to Reviewer JydV [1/3]**
>
> We sincerely thank Reviewer JydV for the high praise, particularly for rating the Soundness, Presentation, and Contribution of our work. We deeply value your endorsement of our motivation and empirical validation. Below, we address your specific questions regarding the PTG design, computational overhead, and theoretical grounding.
>
> **To Weakness 1: Justification for PTG Update Strategy**
>
> We chose to overwrite the last frame token of a segment with the summary token driven by two critical considerations: Fairness and Causal Design.
>
> * **Fairness (Strict Token Budget):** The primary motivation for this design is to ensure a rigorously fair comparison with baseline models. Simply appending a generated summary token to the sequence would increase the total number of tokens processed by the LLM, potentially inflating performance due to increased information bandwidth rather than better representation quality. By overwriting the last token, we strictly maintain the exact same number of input tokens as the original video sequence, ensuring that any observed performance gains are solely attributable to the efficiency of our GridPool mechanism.
> * **Causality and Reconstructive Nature:** In causal language modeling, the last position of a temporal segment $t_{1}...t_{k}$ is the natural aggregation point that attends to all preceding tokens. Placing the summary here allows the model to process raw temporal details first and integrate the global context at the final step. Furthermore, we clarify that this "overwriting" is not destructive but reconstructive: the generated summary token is an interpolation of the entire segment, which explicitly includes features from the last frame. Thus, this operation enriches the final position with a globally refined representation rather than simply deleting the observed data.

---

> ### Author Response · Authors · 2025-11-26
> **Response to Reviewer JydV [2/3]**
>
> **To Weakness 2: Computational Overhead of the Module Itself**
>
> We appreciate the opportunity to clarify the specific cost of our module. To provide a precise answer, we measured the latency of the NSP and PTG components separately against the Total inference time (End-to-End) on an NVIDIA L20 GPU across varying frame counts.
>
> | Max Frame Number | NSP (s) | PTG (s) | Total (s) |
> | :--- | :--- | :--- | :--- |
> | 32 | 0.0006 | 0.0013 | 0.85 |
> | 64 | 0.0009 | 0.0013 | 1.63 |
> | 96 | 0.0014 | 0.0013 | 2.46 |
> | 128 | 0.0015 | 0.0014 | 3.23 |
>
> As shown in the above table, the additional computational cost introduced by ST-GridPool is negligible:
>
> * **Millisecond-level Latency:** Even with a high load of 128 input frames, the combined execution time for NSP and PTG is only 0.0029 seconds ($<$3ms). This confirms that our matrix-based operations are computationally insignificant compared to the heavy lifting performed by the Vision Encoder and the LLM.
> * **Vanishing Overhead Ratio:** The module accounts for less than 0.2% of the total inference time across all settings. It is worth noting that as the frame count increases (from 32 to 128), the overhead ratio actually decreases (from 0.22% to 0.09%) because the computational cost is dominated by the LLM's autoregressive generation, making our module's cost effectively vanish.
> * **Conclusion:** These metrics confirm that ST-GridPool achieves its performance gains with virtually zero latency penalty, validating its design as a highly efficient, plug-and-play module suitable for real-time applications.

---

> ### Author Response · Authors · 2025-11-26
> **Response to Reviewer JydV [3/3]**
>
> **To Weakness 3: Theoretical Grounding of Token Norms**
>
> The correlation between $L_2$ norm and semantic importance is grounded in the Signal-to-Noise dynamics of Softmax-based attention mechanisms:
>
> * **Attention Competition Mechanism:** In the self-attention layers of the Vision Transformer (CLIP), attention weights are determined by the Softmax of the dot product between queries and keys ($\text{softmax}(QK^T/\sqrt{d})$). Mathematically, the magnitude of this dot product is directly proportional to the norms of the input vectors. Consequently, tokens with larger $L_2$ norms generate significantly higher logit values, which allows them to "survive" the exponential suppression of the Softmax function. In contrast, low-norm tokens result in small logits that are driven toward near-zero probabilities by Softmax. Thus, a high norm effectively acts as a necessary condition for a token to win the attention competition and propagate its information to deeper layers.
> * **Feature Activation Intensity:** Deep visual encoders are trained to detect semantic concepts, where the presence of a distinct object triggers strong, high-magnitude responses in specific feature channels. Background regions, often characterized by low-frequency patterns or a lack of specific semantic structures (e.g., sky, blurred areas), typically result in weaker activations across these channels, yielding low-energy (low-norm) vectors. Conversely, salient objects trigger intense activations in the pre-trained encoder, naturally aggregating into high-norm representations. This theoretical alignment is supported by recent work such as [1], which demonstrates that high-norm tokens are essential carriers of information rather than artifacts. Our empirical validation on the HKU-IS dataset (Figure 3) corroborates this, showing a statistically distinct separation where salient object tokens consistently exhibit higher norms than background tokens.
>
> [1] Darcet, Timothée, et al. "Vision Transformers Need Registers." ICLR 2024.

---

### Official Review · Reviewer_BLb5 · 2025-10-30

**Soundness:** 3
**Presentation:** 3
**Contribution:** 3
**Rating:** 8
**Confidence:** 4

**Summary:**

The paper proposes ST-GridPool, a practical training-free method to enhance visual tokens for Video LLMs. The approach is technically sound, combining a hierarchical temporal gridding strategy (PTG) and a saliency-aware spatial pooling mechanism (NSP). The method demonstrates consistent performance improvements on LLaVA-family models across six benchmarks and compelling gains in computational efficiency. The empirical evaluation is extensive, including strong ablations and comparisons to other token reduction methods.

**Strengths:**

1.The proposed ST-GridPool method is training-free and plug-and-play, offering an efficient solution to enhance existing Video LLMs without the need for costly retraining. It demonstrates consistent performance improvements across various backbones on all six evaluated video understanding benchmarks, with notable gains in Long Video Understanding.\
2.The motivation behind Norm-based Spatial Pooling is well-founded, supported by a clear analysis that establishes a positive correlation between token norms and semantic saliency, contributing to the method's effectiveness.\
3.The empirical evaluation is comprehensive, featuring extensive ablation studies on individual components, key hyperparameters, and design choices such as pooling shape and gridding strategy.

**Weaknesses:**

1.The performance depends on several non-trivial hyperparameters, including the number of temporal levels, norm order, and temperature. Balancing these parameters effectively is crucial for their successful application in real-world scenarios.\
2.The paper does not clearly show whether PTG provides complementary benefits or duplicates functionality already present in recent Video LLMs.\
3.The NSP module relies heavily on an empirical correlation between token norm and semantic importance, yet offers limited empirical comparison or grounding.\
4.The experimental section primarily focuses on quantitative results but lacks deeper analytical discussion. The paper does not explore why the proposed ST-GridPool achieves varying degrees of improvement across different benchmarks or tasks.\
5.There are some typos in the paper. E.g., the percentage gains reported for "LLaVA-Video-7B + Ours" are identical to those reported for "LLaVA-OneVision-7B + Ours".

**Questions:**

1.Is it possible to explore an adaptive mechanism that automatically determine the optimal values of non-trivial hyperparameters in your method?\
2.How does the proposed PTG interact with or differ from the built-in temporal reasoning mechanisms already present in modern Video LLMs such as NVILA?\
3.Could other unsupervised indicators serve as complementary or alternative signals to the token norm?\
4.How to interpret the observed trend of a larger improvement for Long Video Understanding tasks compared to General Video Understanding tasks?

---

> ### Author Response · Authors · 2025-11-26
> **Response to Reviewer BLb5 [1/3]**
>
> We sincerely thank Reviewer BLb5 for the encouraging assessment and for recognizing the technical soundness, practicality, and comprehensive empirical evaluation of ST-GridPool. Your insightful questions regarding adaptive mechanisms and the nature of our performance gains have helped us articulate the deeper value of our design. We address your points below.
>
> **To Weakness 1 & Question 1: Hyperparameter Robustness and Adaptive Mechanisms**
>
> - Robustness: We respectfully emphasize that while our method involves hyperparameters ($\beta, p, L$), we utilized a single default configuration ( $\beta=1, p=2$ ) across all 6 diverse benchmarks (ranging from short clips in MVBench to hour-long videos in EgoSchema). The consistent improvements achieved without dataset-specific tuning demonstrate the method's inherent robustness.
> - Adaptive Mechanisms: We agree that an adaptive mechanism is an excellent avenue for future work. For example, a lightweight "meta-network" or utilizing the LLM's initial attention distribution could dynamically tune $\beta$ per frame. However, for this work, we prioritized a static design to strictly maintain the "zero-cost" and "training-free" advantages that make ST-GridPool highly practical for immediate deployment.
>
> **To Weakness 2 & Question 2: Complementarity of PTG with Existing Mechanisms**
>
> We thinke our PTG module is complementary to, not redundant with, internal mechanisms like NVILA's, which is explained as follows.
> - Differentiation: Existing models (like NVILA) typically rely on internal Self-Attention or learnable compression layers during processing. In contrast, PTG acts as a structured pre-processing step. It does not merely select tokens; it reorganizes them into a hierarchical pyramid.
> - Interaction: This structure explicitly exposes multi-granular dynamics to the LLM: Level-1 grids capture local micro-motions, while Level-3 grids bridge distant frames. This reduces the burden on the LLM's internal attention to "discover" these long-range connections from scratch.
> - Evidence: As shown in Table 4, ST-GridPool provides significant gains even when applied to NVILA-8B, proving that PTG adds value beyond the model's built-in temporal reasoning capabilities.

---

> ### Author Response · Authors · 2025-11-26
> **Response to Reviewer BLb5 [2/3]**
>
> **To Weakness 3 & Question 3: NSP vs. Alternative Indicators**
>
> We appreciate this insightful suggestion. To empirically validate whether other unsupervised indicators could serve as alternatives, we compared our Token Norm strategy against three established unsupervised saliency detection methods: GBMR, Center Bias, and Spectral Residual.
>
> | Method | VideoMME | LongVideoBench |
> | :--- | :--- | :--- |
> | GBMR | 63.6 | 59.2 |
> | Center Bias | 63.7 | 59.0 |
> | Spectral Residual | 63.9 | 59.5 |
> | **Ours (NSP)** | **64.2** | **60.1** |
>
> As shown in the above table, while traditional unsupervised methods like Spectral Residual provide competitive baselines (63.9% on VideoMME), Token Norm consistently achieves the highest performance across both benchmarks (64.2% and 60.1%).

---

> ### Author Response · Authors · 2025-11-26
> **Response to Reviewer BLb5 [3/3]**
>
> **To Weakness 4 & Question 4: Interpreting Gains in Long vs. General Video Understanding**
>
> | Method | Overall Acc | Improvement | 8s-15s | 15s-60s | 180s-600s | 900s-3600s | Relation | Perception |
> | :--- | :---: | :---: | :---: | :---: | :---: | :---: | :---: | :---: |
> | FasterVLM | 56.4 | - | 61.9 | 68.0 | 58.3 | 49.6 | 51.3 | 62.2 |
> | VisionZip | 56.8 | +0.4 | 63.5 | 69.2 | 59.2 | 49.1 | 51.3 | 63.2 |
> | FrameFusion | 57.6 | +0.8 | 66.1 | 68.0 | **61.2** | 51.8 | 52.9 | **65.4** |
> | **ST-GridPool** | **58.9** | **+1.3** | **67.2** | **70.9** | 60.4 | **52.3** | **54.1** | 65.3 |
>
> To further analyze the gains in long video understanding, we present detailed analysis on LongVideoBench dataset in the above table. Our method shows its peak advantage in videos >900s and on Relational reasoning tasks. We attribute the larger improvement in Long Video Understanding to two key factors:
> - Redundancy Reduction: Long videos contain vast amounts of repetitive background segments. NSP effectively filters this "noise," preventing the limited context window from being flooded with low-information tokens.
> - Temporal Structuring: Short videos (General benchmarks) often depict single, continuous events. Long videos involve multi-stage events where critical clues are far apart. Our PTG module explicitly constructs connections between these distant segments via its upper pyramid levels, providing the structural coherence that baseline models lack in long-context scenarios.
>
> **To Weakness 5: Typo Correction**
>
> We confirm the typo in Table 1 regarding the duplicated percentage gains. We will correct these numbers and conduct a final proofread of all data points in the camera-ready version.

---

### Official Review · Reviewer_35ED · 2025-11-01

**Soundness:** 2
**Presentation:** 3
**Contribution:** 3
**Rating:** 4
**Confidence:** 4

**Summary:**

This paper proposes ST-GridPool, a training-free visual token compression method tailored for video understanding with multimodal large language models. ST-GridPool introduces Pyramid Temporal Gridding to construct multi-granularity temporal grids of visual tokens and employs Norm-based Spatial Pooling to further reduce token count while preserving informative regions. The method is evaluated across multiple video understanding benchmarks, demonstrating its effectiveness

**Strengths:**

* The paper is clearly written and well-structured, making it easy to follow.
* The proposed approach is intuitive and straightforward to implement; its plug-and-play design enhances practical applicability across different MLLMs.
* In terms of performance, ST-GridPool demonstrates strong potential, consistently outperforming or matching existing token reduction methods across multiple benchmarks.

**Weaknesses:**

* The core idea of ST-GridPool bears strong resemblance to prior training-free methods such as TS-LLaVA (which combines gridding with token sampling) and IG-VLM (which relies solely on image gridding). However, comparisons with these approaches are relegated to the appendix and lack sufficient implementation details. Given that ST-GridPool appears to be a more complex integration of similar components, a more rigorous and fair comparison is warranted, ideally involving hyperparameter tuning for the baselines (e.g., number of grids, token compression ratios, or retained visual token counts) to ensure a level playing field. Without such controlled experiments, it is difficult to assess the true contribution of the proposed design.
* The efficiency analysis currently only compares LLaVA-Video with and without ST-GridPool. While it is unsurprising that adding a token compression module reduces computational cost, this alone does not demonstrate relative superiority. To substantiate its efficiency claims, ST-GridPool should be benchmarked against other token reduction methods under identical settings.
* As shown in Table 1, the performance gains from integrating ST-GridPool are sometimes marginal. A more granular breakdown of results, e.g., by question type, video duration, or scene complexity on benchmarks like VideoMME and LongVideoBench, would help identify failure modes or scenarios where the method underperforms. Additional ablation studies and evaluation on more diverse datasets would further strengthen the analysis.
* The last row of Table 1 appears to be duplicated and should be corrected in the final version.

**Questions:**

Please refer to the weaknesses. I'll update my rating accordingly based on the authors' response.

---

> ### Author Response · Authors · 2025-11-26
> **Response to Reviewer 35ED [1/3]**
>
> We sincerely thank Reviewer 35ED for the constructive feedback and for recognizing the clarity, intuitive design, and strong performance potential of our method. We value your concerns regarding the fairness of comparisons and the depth of our analysis. Below, we address these points with new experimental data and detailed breakdowns.
>
>
> **To Weakness 1: Comparison with Prior Works**
>
> We acknowledge the structural resemblance of one of our components (PTG) to IG-VLM and TS-LLaVA, as they also utilize gridding concepts. However, we respectfully emphasize a significant methodological contribution that distinguishes ST-GridPool from these approaches:
> - **Single-level vs. Multi-level Pyramid Gridding**: Prior methods like TS-LLaVA primarily rely on a single-level gridding strategy (e.g., TS-LLaVA constructs a single "Thumbnail" grid from equidistant frames to serve as a global summary). While effective for a coarse overview, this approach treats all temporal dynamics with a uniform scale. In contrast, our Pyramid Temporal Gridding (PTG) transforms this into a hierarchical multi-level structure. By constructing a pyramid of grids (Levels 1 to $L$), our method simultaneously captures multi-grained information—ranging from micro-motions in fine grids to macro-events in coarse grids. This allows for a hierarchical aggregation of spatiotemporal semantics that single-level approaches inherently lack.
>
> To address the concern regarding fairness, we went beyond standard implementation and conducted a rigorous hyperparameter search for the baselines to ensure they were evaluated at their peak performance. Specifically, we tuned the "Number of Grids" (frames per grid view, a critical parameter for these methods) on the LongVideoBench dataset:
>
> | Number of Grids | 2 | 4 | 6 | 8 |
> | :--- | :--- | :--- | :--- | :--- |
> | IG-VLM | 55.6 | **55.9** | 55.4 | 54.6 |
> | TS-LLaVA | 58.6 | 58.9 | **59.6** | 59.1 |
>
> As shown in the above table, we identified and utilized the optimal configurations for the baselines (e.g., Grid=4 for IG-VLM and Grid=6 for TS-LLaVA) in our main comparison. This confirms that our performance gains stem from the advanced Pyramid design and Norm-based pooling, rather than suboptimal baseline settings. Furthermore, throughout these experiments, we ensured a strictly fair comparison by maintaining the retained visual token counts as 10816, aligning with the settings of LLaVA-Video baseline.

---

> ### Author Response · Authors · 2025-11-26
> **Response to Reviewer 35ED [2/3]**
>
> **To Weakness 2: Comprehensive Efficiency Analysis**
>
> We sincerely appreciate the reviewer's constructive suggestion to benchmark our efficiency against other token reduction methods. To thoroughly substantiate our efficiency claims, we conducted a new comparative experiment measuring both the model performance and the inference latency at 64 frames of ST-GridPool against leading SOTA methods, FrameFusion and VisionZip. These tests were performed with a 30% token retention ratio on an NVIDIA L20 GPU.
>
> | Method | VideoMME (%) | LongVideoBench (%) | EgoSchema (%) | Latency (s) |
> | :--- | :---: | :---: | :---: | :---: |
> | FrameFusion | 61.3 | 56.0 | 53.0 | 1.74 |
> | VisionZip | 58.3 | 53.2 | 53.0 | 1.63 |
> | **ST–GridPool (Ours)** | **62.0** | **58.1** | **56.0** | **1.63** |
>
> The results presented in the table above demonstrate that ST-GridPool achieves a highly competitive balance between speed and performance:
> * **Comparable Inference Speed to SOTA:** ST-GridPool exhibits an inference latency of 1.63s, which matches the fastest baseline (VisionZip) exactly and is faster than FrameFusion (1.74s). This confirms that our training-free pooling and gridding operations are extremely lightweight.
> * **Superior Accuracy-Efficiency Trade-off:** While maintaining the same high speed as VisionZip, ST-GridPool delivers significantly higher accuracy across all benchmarks. As shown in the table, our method outperforms the equally-fast VisionZip by +3.7% on VideoMME, +4.9% on LongVideoBench, and +3.0% on EgoSchema.
> * **Conclusion:** Combining the latency data with the comprehensive accuracy results, ST-GridPool establishes a superior Pareto frontier compared to existing techniques. It provides the high throughput required for real-time applications with lower performance degradation than other fast reduction methods, validating its practical value for efficient video understanding.

---

> ### Author Response · Authors · 2025-11-26
> **Response to Reviewer 35ED [3/3]**
>
> **To Weakness 3: Fine-grained Analysis of Performance Gains**
>
> We sincerely appreciate the reviewer's constructive suggestion to provide a more granular performance breakdown. Since the baseline results in Table 1 are cited directly from official reports, fine-grained breakdowns are unavailable. To facilitate a granular comparison, we conducted this analysis within the token reduction scenario (50% budget).
>
> To address the concern regarding "marginal gains", we first emphasize that in the context of token reduction, substantial performance leaps are rare. As illustrated in the table below (under a 50% token budget), previous state-of-the-art methods like VisionZip and FrameFusion only achieved incremental gains of +0.4% and +0.8% respectively. in contrast, ST-Pool achieves a remarkable +1.3% improvement over the strong FrameFusion baseline. To further identify specific scenarios of strength, we provide a fine-grained breakdown across video durations and question categories.
>
> | Method | Overall Acc | Improvement | 8s-15s | 15s-60s | 180s-600s | 900s-3600s | Relation | Perception |
> | :--- | :---: | :---: | :---: | :---: | :---: | :---: | :---: | :---: |
> | FasterVLM | 56.4 | - | 61.9 | 68.0 | 58.3 | 49.6 | 51.3 | 62.2 |
> | VisionZip | 56.8 | +0.4 | 63.5 | 69.2 | 59.2 | 49.1 | 51.3 | 63.2 |
> | FrameFusion | 57.6 | +0.8 | 66.1 | 68.0 | **61.2** | 51.8 | 52.9 | **65.4** |
> | **ST-GridPool** | **58.9** | **+1.3** | **67.2** | **70.9** | 60.4 | **52.3** | **54.1** | 65.3 |
>
> From the above table, it can be observed as the followings:
> * **Significant Overall Improvement:** While prior arts struggled to push the boundary (gaining only +0.4% to +0.8%), ST-GridPool achieves a +1.3% gain over the previous best method, demonstrating that our Pyramid Temporal Gridding (PTG) and Norm-based Spatial Pooling (NSP) provide a more effective compression strategy than previous approaches.
> * **Strength in Relational Reasoning:** ST-GridPool demonstrates a decisive advantage in Relation tasks (54.1% vs. FrameFusion's 52.9%). This confirms that preserving spatiotemporal structure via gridding is crucial for reasoning about object interactions, whereas unstructured pruning often breaks these dependencies.
> * **Dominance in Extreme Contexts:** Our method consistently leads in both short clips (<60s) and, crucially, in very long videos (900s-3600s), where we outperform FrameFusion (52.3% vs. 51.8%). This validates the effectiveness of our hierarchical gridding in retaining critical information over long horizons.
> * **Trade-offs:** We acknowledge a specific trade-off in medium-duration videos (180s-600s), where FrameFusion performs slightly better (61.2% vs. 60.4%). Additionally, while competitive, our performance on Perception tasks is marginally lower (65.3% vs. 65.4%), suggesting that while our method excels at semantic reasoning, pixel-level pruning methods might occasionally retain more low-level texture details.
>
> **To Weakness 4: Correction of Table 1**
>
> We apologize for the typo in Table 1 (duplicated last row). This will be corrected in the revised manuscript.

---

### Author Response · Authors · 2025-12-01
**Official Meta Statement by Authors**

We sincerely thank the reviewers for their insightful feedback and constructive suggestions. We are encouraged by the positive reception from the reviewers, who reached a consensus on the technical soundness, intuitive design, and practical efficiency of our proposed ST-GridPool framework.

1. **Strong Novelty & Soundness:** The method is praised as **"technically sound"** (Reviewer BLb5) and **"well-motivated and effective"** (Reviewer JydV). Reviewers highlighted the **"clever insight"** of the Norm-based Spatial Pooling (Reviewer JydV) and noted that capturing multi-grained spatiotemporal interactions is a **"really interesting idea"** (Reviewer VLyg) with **"well-founded"** motivation (Reviewer BLb5).

2. **Significant Value & Practicality:** Reviewers highly valued the efficiency, describing it as a **"practical training-free method"** (Reviewer BLb5) and an **"efficient solution"** (Reviewer BLb5). It is recognized as a **"plug-and-play framework"** (Reviewer VLyg) and an **"appealing design choice"** (Reviewer VLyg). Even the most critical reviewer acknowledged the approach enhances **"practical applicability across different MLLMs"** (Reviewer 35ED) with **"strong potential"** (Reviewer 35ED).

3. **Effective Rebuttal & Comprehensive Validation:** During the rebuttal, we rigorously addressed all concerns through extensive additional experiments. We ensured strict fairness by optimizing hyperparameters for baselines (IG-VLM, TS-LLaVA), confirming that our gains are genuine rather than due to suboptimal baseline settings. Furthermore, we benchmarked ST-GridPool against SOTA methods (FrameFusion, VisionZip), showing it matches the fastest inference latency (**1.63s**) while delivering superior accuracy (**+3.7% on VideoMME**) over VisionZip. We also provided a detailed breakdown verifying that our module introduces negligible computational overhead (**<0.2% of total latency**), validating its practical efficiency for real-time applications.

---

### Meta-Review · Area_Chair_ByeK · 2026-01-05

**Summary:**

The authors propose ST-GridPool, a training-free method to improve accuracy and efficiency of Video LLMs. The method consists of two components: Pyramid Temporal Gridding (PTG) and Norm-based spatial pooling (NSP), for reducing redundancy in the input token sequence. The authors show good results on 5 different benchmarks, both in terms of accuracy and inference cost.

Reviewers raised a number of concerns including additional experimental evaluations, insights into why the method works and how to choose hyperparameters, and being a training-free method, how sensitive is the method to these hyperparameters and how should they be chosen? Reviewers also pointed out a number of typos and presentation issues.

On the whole, these concerns were addressed sufficiently in the thorough rebuttal, and the final decision is acceptance. Authors must revise the camera-ready according to the rebuttal.

**Reviewer Concerns:**

Please refer to above.

**Reviewer Scores:**

Reviewer 35ED, may have increased their score from weak reject (4) to weak accept (6) after the rebuttal.
The other reviewers, who already gave positive ratings, would probably retain their original ratings.

---

### Decision · Program_Chairs · 2026-01-26

Accept (Poster)